# Interplay between Nutrition and Hearing Loss: State of Art

**DOI:** 10.3390/nu11010035

**Published:** 2018-12-24

**Authors:** Ana M. Puga, María A. Pajares, Gregorio Varela-Moreiras, Teresa Partearroyo

**Affiliations:** 1Department of Pharmaceutical and Health Sciences, Faculty of Pharmacy, CEU San Pablo University, 28668 Madrid, Spain; anamaria.pugagimenezazca@ceu.es (A.M.P.); gvarela@ceu.es (G.V.-M.); 2Department of Structural and Chemical Biology, Centro de Investigaciones Biológicas (CSIC), 28040 Madrid, Spain; mapajares@cib.csic.es; 3Molecular Hepatology Group, Hospital La Paz Institute for Health Research (IdiPAZ), 28046 Madrid, Spain

**Keywords:** auditory function, presbycusis, noise induced hearing loss, caloric restriction, proteins, lipids, carbohydrates, vitamins, minerals, antioxidants

## Abstract

Hearing loss has been recently ranked as the fifth leading cause of years lived with disability, ahead of many other chronic diseases such as diabetes, dementia, or chronic obstructive pulmonary disease. Moreover, according to the World Health Organization, moderate-to-profound hearing loss affects about 466 million people worldwide. Its incidence varies in each population segment, affecting approximately 10% of children and increasing to 30% of the population over 65 years. However, hearing loss receives still very limited research funding and public awareness. This sensory impairment is caused by genetic and environmental factors, and among the latter, the nutritional status has acquired relevance due its association to hearing loss detected in recent epidemiological studies. Several experimental models have proved that the onset and progression of hearing loss are closely linked to the availability of nutrients and their metabolism. Here, we have reviewed studies focused on nutrient effects on auditory function. These studies support the potential of nutritional therapy for the protection against hearing loss progression, which is especially relevant to the aging process and related quality of life.

## 1. Introduction

Hearing loss (HL) is a common disorder that has multifactorial origin, including both genetic and environmental factors [1,2]. Genetic factors include mutations in genes or regulatory elements involved in the development, structure, or function of the cochlea. Among environmental factors, exposure to noise, the increasing use of listening devices, ototoxic drugs (i.e., antibiotics, anticancer drugs, pain killers), or nutritional deficiencies can be listed. Therefore, this is one of the fields in which nutritional intervention studies may have a greater preventive potential, especially when HL is also associated with age [3]. According to the World Health Organization (WHO) around 466 million people worldwide (over 5% of the world’s population) suffer from disabling HL [4]. Its incidence varies in each population segment, ranging from approximately 10% of children to 35% of the population over 65 years [2,5]. Moreover, it is estimated that only in Europe, the number of HL cases will increase more than 18% in the 2010–2020 decade [6]. In fact, estimations indicate that over 900 million people worldwide will have disabling HL by 2050 [4]. Worsening of the situation is highlighted by the fact that HL has moved seven positions towards the top of the ranking of causes of years lived with disability since 2010, reaching now the fourth position and placing itself ahead of headline-grabbing conditions such as diabetes or dementia [7,8]. 

HL has important consequences in the quality of people’s lives [9]. The reported effects include: (i) emotional reactions such as loneliness, isolation, depression, anxiety, or frustration; (ii) behavioral reactions namely blaming, withdrawing, or bluffing; and (iii) cognitive reactions including confusion, distracting thoughts, decreased self-esteem, and communication disorders [8,10,11]. The final consequence of these effects is a significant increase in the risk of dependence and associated costs. In a recently published report, the WHO estimates the annual cost of unaddressed HL to be globally about 750 billion US dollars [12].

More than twenty-two active randomized clinical drug trials are being conducted currently in the United States of America (USA), some of them with N-acetilcysteine, vestipitant, zonisamide, anakinra, sodium thiosulfate, alpha-lipoic acid, or D-methionine and six potential therapeutic molecules are under research [13]. Interestingly, it is well known that the prevalence and impact of HL can be mitigated through public health actions. As an example, recent estimations suggest that virtually 60% of HL among children can be prevented through public health measures [3] and around 50% with immunizations for rubella, mumps, measles, and meningitis [7]. Despite the importance of early diagnosis and appropriate intervention in HL, only a few countries, mainly in the high-income group, have implemented strategic plans to address this injury [14]. Different epidemiological studies worldwide have demonstrated the association between the deficiency of several essential nutrients and HL [15,16,17,18], while others also provided evidences of its prevention by means of dietary supplementation [19,20,21]. Moreover, dietary exposure to potential ototoxic heavy metals such as cadmium and lead [22], obesity (measured as high body mass index and high waist circumference), and reduced physical activity are also related to HL, confirming the negative consequences of an unhealthy diet and lifestyle in the auditory function [9]. Therefore, the aim of this review is to summarize the current knowledge on the effects of dietary habits into the auditory function, the impact of micronutrient deficiencies or insufficiencies and the potential of nutritional therapies in the prevention of HL progression. For a better understanding and follow-up, we have structured this review based on the effects of caloric restriction, macronutrients, micronutrients and other nutrients on auditory function and HL.

## 2. Anatomy of the Ear and Hearing Loss

For a better comprehension of the present review, a slight description of the ear anatomy is included below. The ear is divided into three anatomical compartments: outer, middle and inner ear (Figure 1A). The inner part of the ear contains the cochlea (Figure 1B), responsible for the mechanotransduction of the sound stimulus. The cochlear duct (Figure 1B) has a section with triangular prism shape (scala media) composed by: (i) an external wall, the spiral ligament and stria vascularis; (ii) an upper wall, the Reissner membrane; and (iii) a lower part, the basilar membrane on which the organ of Corti is supported (Figure 1C) [23]. The organ of Corti is a specialized epithelium, which acts as hearing receptor and contains two types of hair cells, outer and inner, innervated by the spiral ganglion neurons that connect it with the brain [23,24].

There are different criteria to classify HL. According to the location of the injury, HL is classified as: (i) conductive, when the damage occurs either in the middle or the external ear; (ii) sensorineural HL (SNHL), if the injury affects the inner ear (usually the cochlea) or the auditory nervous system; and (iii) mixed, a combination of both [11]. Regarding its severity, the American Speech–Language–Hearing Association establishes different degrees of HL depending on the range of decibels of hearing level (dB HL) that are lost: (i) normal (−10–15 dB HL); (ii) slight (16–25 dB HL); (iii) mild (26–40 dB HL); (iv) moderate (41–55 dB HL); (v) moderately severe (56–70 dB HL); (vi) severe (71–90 dB HL); and (vii) profound (above 91 dB HL) [27]. Considering the age at which the damage appears, HL is classified as: (i) prelocutive or prelingual, appearing prior to the acquisition of language; and (ii) postlocutive or postlingual, arising later on [28]. Finally, HL can be non-syndromic if it is the only dysfunction present or syndromic when associated with alterations in other organs.

Pure tone audiometry is one of the most commonly used methods for assessment of the auditory function. It measures specific frequency thresholds and informs on the conductive or sensorineural origin of the detected HL. However, in recent years different and more objective methods to evaluate the hearing function have been developed, which avoid the high dependence of audiometry on patient’s cooperation [29]. Among them, assessment of the auditory brain response (ABR), a non-invasive technique based on the sound-evoked potentials, allows the determination of an objective auditory threshold. Moreover, ABR provides information about the neural transmission conduction through the peripheral and brainstem portions of the auditory pathway [30]. This method is generally used to assess hearing in animal models, although it requires anesthesia.

Different epidemiological studies allowed the identification of HL risk factors and their classification as modifiable and non-modifiable risk factors. Generally, risk factors are classified in 4 different categories: (i) aging; (ii) environmental (occupational and leisure noise exposure [31], treatment with ototoxic drugs [32], nutritional deficiencies [18], smoking and adiposity [11], and even the socioeconomic status); (iii) genetic predisposing (sex, race, or specific genes); and (iv) health co-morbidities (diabetes, hypertension, stroke, and/or smoking) [11,33,34]. More than 300 genes have been associated with hearing [35] and out of them single nucleotide polymorphisms in more than 50 genes have been related to HL [36]. Consistent associations have been established with some non-modifiable risk factors, such as increased age or male ethnic background. For example, Agrawal et al. observed that White and Mexican American men had higher prevalence of bilateral and high-frequency HL, whereas African-American men showed the lowest prevalence among the ethnic groups studied [37]. In any case, lifestyle is a key factor that may influence not only the auditory performance at any age, but also the progression of HL in the elderly, also known as presbycusis or age-related HL (ARHL) [38]. Examples of the influence of life style are diverse. Among them, the increased risk of HL detected in smokers and passive smokers that is dose-dependent [39], the better hearing detected in the elderly with moderate alcohol consumption versus heavy alcohol drinkers, which had a leaning toward more pronounced HL, or the observation that a high body mass index is related to an increased risk to develop HL at high frequencies [40]. 

The functional decline of the organism during aging is usually associated with a progressive sensory injury that is often concomitant with cognitive decline [41,42], but also with the development of a great variety of chronic illnesses such as cancer, diabetes, atherosclerosis, osteoporosis, and cardiovascular diseases. HL shares common biochemical alterations with some of these chronic diseases that lead to cellular degeneration and organ malfunction [42]. Among these alterations, increased inflammation, impairment of insulin signaling, abnormal proteostasis, oxidative stress and alterations in the intermediary metabolism of the inner ear have been reported [43]. As a consequence, the percentage of elderly people affected by presbycusis increases with age from 30% in people aged more than 65 to about 80% in those older than 85 years [44]. 

Presbycusis is a degenerative disorder identified as SNHL, since the dysfunction arises in the cochlea, specifically in the organ of Corti. Sound-induced vibrations are transduced by sensory hair cells of the cochlea into electrical signals, information that cochlear neurons communicate to the brain [23,45]. Both stria vascularis and hair cells are particularly susceptible to damage; hence, their affectation leads to a decline in hearing function. HL in presbycusis usually begins in the high-frequency regions, compromising the understanding of conversations in noisy environments and tends to spread toward the low-frequency region during aging [46]. If it reaches the range of 2–4 kHz, important for distinguishing deaf or voiceless consonants (those produced without sound from the vocal cords, such as f, k, p, s, or t), the understanding of conversations in any situation is affected. HL progresses by approximately 1dB per year, once 60 years of age are reached [47]. Among the injuries causing ARHL, age-related loss of hair cells within the stria, oxidative stress (from noise exposure or other causes of free radical production) or microvascular disease of strial vessels should be mentioned [34]. Thus, targeting these traits may offer interesting opportunities to slow down the progression of HL (e.g., using antioxidants) [48,49]. The search for a suitable treatment of these disorders, not only in the elderly but at all stages of life, represents a key health challenge and a major socioeconomic problem [9], especially since the population above 60 is expected to double by 2050 in the European Union [50]. 

## 3. Caloric Restriction and Hearing loss

It is well known that one of the most successful strategies to retard the negative impact of aging in the organism is to reduce dietary intake through caloric (CR) and/or dietary restriction (DR); the latter is defined as a reduction in the calorie intake with adequate nutrient levels [51]. In different animal models (rodents and non-human primates), CR has been shown to decrease the incidence of obesity, diabetes and tumors [51,52,53]. Moreover, CR can protect neurons in animal models of neurodegenerative diseases [54] and decreases the levels of oxidative protein damage in aged brains, hearts and livers [55]. In humans, research of the anti-aging effects of CR is still at an early stage, although it has been demonstrated that CR reduces the incidence of obesity, as well as cholesterol levels, blood pressure, oxidative stress, inflammation, and insulin resistance [53,55]. The precise mechanisms by which CR/DR maintain a good health and prolong life is not completely understood, but apparently involve numerous adaptations of cellular metabolism and energy production [56]. One of the most accepted theories is the Mitochondrial Free Radical Theory of Aging, which postulates that aging and age-related diseases result from the accumulation of oxidative damage caused by reactive oxygen species (ROS) originated in the mitochondria. In this context, the major antioxidant defense is carried out by the glutathione and thioredoxin systems. Antiaging effects of CR seem to be associated with the mitochondrial antioxidant defense pathway, but the exact mechanism remains unclear [55]. Although these considerations apply to the general health state, several researchers have tried to elucidate if CR/DR is able to counteract the effects of aging in HL using animal models (Table 1). For example, Mannström at al. performed a detailed evaluation of the impact of DR on the total number of sensory hair cells and spiral ganglion neurons and on the volume and the fine structure of the stria vascularis in Sprague–Dawley albino rats [57]. Interestingly, these rats present high similarities to humans in terms of life span and sensory and motor functions during aging. Comparison of results obtained in animals under DR versus rats fed ad libitum demonstrated that DR maintained the auditory reflex, as well as the cellular integrity of the stria vascularis. 

To understand the biochemical mechanisms of the antiaging effect of CR, the role of sirtuins, a group of NAD^+^-dependent protein deacetylases, has been widely studied. Specifically, mitochondrial sirtuin-3 (SIRT3) has the capacity to regulate ATP levels and the activity of complex I of the electron transport chain, thus playing an important role in the metabolic reprogramming mediated by CR [46]. Several studies demonstrated the effect of CR on SIRT3 levels in different settings. As an example, fasting increases SIRT3 protein expression in liver mitochondria [58] or primary mouse cardiomyocytes [59] providing protection against oxidative-induced cell death. Regarding the effect of SIRT3, Someya et al. proved that CR prevents ARHL in C57BL/6J mice, reducing degeneration and inducing *Sirt3* in the cochlea, using regular and CR (75% reduction in caloric intake) diets to feed wild-type and *Sirt3*-deficient mice [60]. CR delayed the progression of HL in wild-type mice, but not in *Sirt3*-deficient animals, demonstrating the critical role of this sirtuin in the effect of CR in ARHL. Then, the metabolic effects of CR were studied in both models. The findings indicate lower levels of serum insulin and triglycerides in wild-type, but not in *Sirt3*-deficient mice, leading the authors to conclude that these results are a consequence of the SIRT3 role in metabolic adaptations to CR. Subsequent studies demonstrated that SIRT3 enhances the glutathione antioxidant defense system and decreases ROS levels in mammals under CR conditions, resulting in protection of the inner ear and preventing ARHL [46]. Given the mitochondrial location of this protein, it was postulated that SIRT3 could regulate any of the mitochondrial antioxidant systems. Hence, Someya et al. observed that SIRT3 is able to modulate the reduction of oxidized glutathione after CR. Under these conditions, SIRT3 binds and deacetylates mitochondrial isocitrate dehydrogenase 2 (IDH2), increasing its enzymatic activity. Hence, it was proposed that CR promotes *Sirt3* expression, leading to deacetylation and activation of IDH2 and, in turn increasing the resistance to oxidative stress and preventing ARHL [46]. Interestingly, these authors postulate that this is the main mechanism of aging retardation by CR and that pharmaceutical interventions that induce SIRT3 activity will mimic this diet by increasing resistance against oxidative stress and preventing the mitochondrial decline associated with aging. Although anti-aging effects of CR were thought to require a significant reduction of body weight, the study carried out by Someya et al. [46], revealed that despite the significant reduction of body weight in *Sirt3* knockout mice, CR failed to slow down the development of ARHL. Therefore, an essential role of specific proteins such as SIRT3 was established, together with the conclusion that weight loss alone is not enough for the preservation of normal hearing.

Studies to evaluate the effect of CR in auditory function have been also performed in additional animal models. Rhesus monkeys (*Macaca mulatta*) are long-lived primates, phylogenetically closer to humans than rodents, and hence constitute a better model to study the effects of CR in human ARHL. Their maximum lifespan is 40 years and they display a decrease in the auditory function by 25–31 years of age [61]. Moreover, their auditory structures (middle ear, cochlea) and neural function are very similar to those of humans. The University of Wisconsin (UW) study was the first to assess the effect of CR in hearing with a large cohort of animals, using a problem group that was maintained with 30% less calories than the control group for 3 to 9 years, when the animals aged to between 11 and 23-years old. Results of the auditory tests showed a tendency of ABR thresholds of male monkeys under CR to be lower (better auditory function) than those in the control group (no significant differences) [62,63]. Torre et al. evaluated the effect of CR in ARHL comparing monkeys fed a low fat, high fiber diet ad libitum with others fed a 30% lower amount of the same diet [64]. Studies of the auditory function performed after 12–13 years on these diets (mean ages 18.7–20.4 years), revealed no differences between groups. Finally, other study compared animals fed a control diet versus a diet with 70% less calories, but supplemented with vitamins and minerals, after a follow up of 8–13.5 years. Auditory tests were performed at a mean age of 20.5 and 20.6 years for the control and CR groups, respectively, and no effects in auditory function were found despite the large energy reduction due to this CR diet [65]. The authors concluded that the duration of CR is a critical factor to observe its protective effects on ARHL, especially considering the long lifespan of *Rhesus macaque*. Moreover, it is important to consider the age of the animals at the time of the auditory tests. A close analysis of these three studies indicates that monkeys were still relatively young and had not reached the age at which ARHL is normally shown in the auditory tests. Therefore, it would be interesting to evaluate the auditory function of the animals at more advanced ages, to confirm whether CR has any protective effect in the auditory function. 

## 4. Macronutrients and Hearing Loss

Among macronutrients, the limited research carried out to analyze the impact of carbohydrates and proteins on auditory function and the prevention of HL suggests that they possibly have marginal effects on hearing (Table 2). In fact, the relationship between carbohydrate consumption and HL is thought to be related, not to carbohydrates themselves, but to the triglyceride (TG) serum levels. In this line, diets rich in carbohydrates, especially sugars and concentrated sweets that are highly concentrated in fructose, led to high TG serum levels, and hence, expected to affect auditory function [66,67]. In fact, Gopinath et al. observed that, human adults with high glycemic index, higher glycemic load and overall higher total carbohydrate levels showed an increased risk of HL [68]. In view of the results and in agreement with the opinion of the authors, high postprandial glycemia might be a potential underlying mechanism for developing ARHL.

Regarding the role of proteins in HL, so far, the available studies have been mainly focused on the role of specific proteins such as β-Conglycinin, one of the major storage proteins of soy. Initial studies performed to analyze the protective effect of this protein against ARHL demonstrated that foods containing β-Conglycinin are useful for the prevention of obesity and dislipemia [69]. Tanigawa et al. then used six-month-old male C57BL/6J mice that were fed either β-Conglycinin (study group) or casein (control group) for 6 months [70]. At different time points, ABR tests, analysis of the cochlear blood flow and histological studies were performed. High β-Conglycinin ingestion prevented the increase of ABR thresholds shown in the control group, preserving the cochlear blood flow and improving the oxidative status [70]. To the best of our knowledge, no epidemiological studies have been carried out to evaluate the effect of this protein in HL, and hence to assess whether the potential beneficial effects observed in animals also apply to humans.

Conversely, numerous and detailed research on the effect of lipid consumption in the auditory function has been carried out (Table 2). The interest on the study of lipid effects on the auditory function is based on the observation that, in the Mabaan tribe, there were no cases of cardiovascular disease or AHRL and that additional studies supported a correlation between cardiovascular events and HL [71,72]. In this context, the connection between *n*-3 polyunsaturated fatty acids (also called ω-3 fatty acids (ω-3)) and HL was suggested by the role of these lipids in the prevention/protection against vascular diseases. Briefly, ω-3 belong to the 18 to 24 carbon family with three or more double bonds, the last of these bonds being located three carbons away from the terminal methyl group (starting at the carboxyl group). Mammals do not have the enzymatic ability to create double bonds beyond the ninth carbon, and so ω-3 must be obtained from the diet. The simplest ω-3 is α-linolenic acid (ALA), which is found in certain vegetable oils (flaxseed, canola and soybean) and walnuts. ALA is the metabolic precursor of eicosapentaenoic acid (EPA) and docosahexaenoic acid (DHA), but given the limited capacity of mammals for their synthesis, both must be also provided by the diet, mainly from fish and fish oil. Their beneficial vascular effects are related to a variety of actions, including their hypolipidemic, anti-inflammatory and anti-atherotrombotic properties and their capacity to lower TG levels. EPA, DHA, and ALA are candidate molecules for cardiovascular disease prevention [73], although differences among them have been shown. For example, EPA is known to be more effective than DHA in reducing inflammation [74], therefore it could be expected to be more effective in the prevention of HL. The relationship between between cardiovascular disease, risk factors and auditory status was studied by Gates et al. in the Framingham cohort [75], observing a moderate association between cardiovascular events (including ischemic heart disease, myocardial infarction and stroke) and HL. In parallel, different animal models have given insights into the relationship between AHRL and vascular events. As an example, gerbils present progressive reduction of the blood flow and microvascular alterations in the stria vascularis and spiral ligament, which are related to HL and aging [76]. These data suggested that, because of the decreased blood flow, the cochlear supply of essential nutrients (including ω-3) is reduced, leading to metabolic alterations and triggering the progression of HL. 

A preliminary epidemiological study was carried out in Finland in the 1970s in two different psychiatric hospitals to evaluate the effect of different dietary lipids in the auditory function. During a 5-year period, one hospital maintained the usual high saturated fat diet, whereas it was substituted by high polyunsaturated fat in the other. At the end of this period, the incidence of coronary problems and the number of people showing auditory impairment were significantly lower in the second hospital. Then, diets of both hospitals were interchanged and four years later, the inverse results were observed. Thus, it was established that auditory function improves with a diet rich in polyunsaturated lipids, whereas it deteriorates with high-saturated fat; the incidence of coronary diseases followed the same pattern. The Finnish investigators concluded that the diet is a critical factor to prevent coronary heart diseases and that it may well stop or even reverse HL [19]. Some years later, a cross-sectional and 3-year longitudinal study was carried out by Dullemeijer et al. to investigate if plasmatic levels of long chain ω-3 were associated with ARHL [20]. Volunteers were classified into quartiles according to their plasmatic proportions of long chain ω-3. Results showed a better auditory function at low frequencies in volunteers belonging to the highest quartile of plasma long chain ω-3 as compared to those in the lowest quartile over the three years of study. However, plasmatic levels of ω-3 showed no association with HL in the high frequencies. These differences were ascribed to the fact that the apex of the cochlea transduces the low-frequency sounds, whereas the base transduces high frequency ones. Moreover, blood supply lies farther away from the apex than from the base. Thus, in the event of vascular disease, the apex may be more prone to changes in microcirculation, resulting in alterations of the auditory function at low frequencies. Conversely, improvements in microcirculation caused by high plasma long chain ω-3 levels may be associated to decreases of low-frequency thresholds [20]. Despite its interest, limitations of this study include that half of the volunteers received folic acid (FA) supplementation for three years, due to their participation in a randomized controlled trial. Since FA is known to slow down the decline of HL in low frequencies [77], a certain bias in the results might be expected. 

Some other studies have focused on the effect of fish consumption, rich in ω-3, in the auditory function. A prospective Australian study carried out in women evaluated the association between dietary intake of ω-3 from fish and the risk of presbycusis, finding that high dietary intake of these lipids was associated with a 24% decrease in the risk of developing HL. Researchers observed as well that regular ingestion of fish (>1 but <2 servings/week) was negatively associated with the 5-years incidence and progression of HL in older adults, but higher fish consumption (>2 servings/week) did not protect against its progression. These findings suggested the existence of a threshold of the beneficial effect at 1–2 servings of fish/week. However, when fatty acids were analyzed individually, it was not possible to establish an association between dietary intake of ALA and the incidence of HL, possibly due to its poor conversion (<5%) to EPA and DHA [73]. Finally, as observed in previously mentioned studies, significant inverse association between dietary total long chain ω-3 in plasma and HL prevalence was found [21]. In general, these results confirmed the potential of changes in the nutritional status of older adults as a possible strategy to diminish the public burden of ARLH [21]. The effect of ω-3 consumption was studied not only for presbycusis, but also for HL in general population. Curhan et al. prospectively examined the relationship of total fish intake, consumption of specific fish types, ingestion of long-chain ω-3 and the risk of self-reported HL in a cohort of 65,215 American women aged 27–44, who were followed for 18 years [78]. Results of this study showed that women who consumed 2 or more fish servings per week, independently of the fish type (tuna, dark-meat fish, light-meat fish, or shellfish) presented a reduction in HL and, additionally, the higher intake of long-chain ω-3 were inversely associated with risk. Although these results were consistent with those of the Australian study previously explained, differed from data of another cross-sectional European study in which no association between fish intake and hearing levels was found in women [79]. The authors concluded that these differences rely in the small population analyzed and the cross-sectional design of the European study that prevented the establishment of temporal relationships. 

In contrast, animal studies rendered different results. Church et al. used Wistar rats to evaluate the putative effects that ω-3 ingestion during pregnancy and lactation have in the offspring lifespan, brain and sensory development and function (evaluated using ABR) [80]. In this study, the animals were fed three different diets from the first day of pregnancy until end of lactation: control, ω-3 deficient and high ω-3 diets providing ω-3/ ω-6 ratios ranging from 0.14 to 14. Permanent raises in ABR thresholds, i.e., HL, in the offspring from rats fed an excess of ω-3 were detected, together with abnormally large ABR amplitudes in response to high stimulus intensities (i.e., hyperacusis), due to compromised neural inhibition. It was hypothesized that these adverse effects could be caused by several mechanisms, among which two can be highlighted: (i) alterations in cell membrane, organ, brain and sensory function due to excessive ω-3 plus low arachidonic acid levels in blood; and (ii) the oxidative stress and subsequent cell apoptosis induced by the high ingestion of ω-3. Thus, the authors emphasized that ω-3 excess or a high ω-3/ω-6 ratio during pregnancy and lactation, as well as in infant formula feeding could result in sensory disorders [80]. 

In addition to the role of ω-3 in cardiovascular disease and HL, these lipids show also beneficial effects in inflammation and oxidative stress. Furthermore, high plasmatic levels of homocysteine (Hcy) are an independent risk factor for cardiovascular and HL and several epidemiologic studies have shown opposite effects of fish oil supplementation [81,82]. These data prompted Martínez-Vega et al. to perform a study focused on the putative benefit of an ω-3 supplemented diet on cochlear Hcy metabolism using a classical model of early HL, the C57BL/6J mice. Two experimental groups, which were fed control or ω-3-supplemeted diets *ad libitum* for 8 months, were analyzed. The results obtained showed that the ω-3-supplemeted diet improved HL and led to changes in the cochlear Hcy metabolism and to modifications in cytokine levels. Likewise, this study revealed that high ω-3-supplemeted diet prevented changes in the expression of proinflammatory cytokines, and *Bhmt* and *Cbs* genes (both involved in Hcy metabolism). Furthermore, there was an increase in BHMT, an enzyme involved in remethylation of Hcy for the conservation of methionine levels [83]. This cluster of results confirmed the close relationship between Hcy and ω-3 metabolism and consequently with HL. In the vitamins section of this review the role of FA, whose deficit leads to elevations in plasmatic Hcy levels, in HL will be deeply analyzed.

Many other researchers focused not on the potential beneficial effects of ω-3, but on the detrimental association between dietary intake of fats (saturated and cholesterol) and HL. Studies using different animal models (guinea pigs and chinchillas) showed an increase of hearing thresholds when feeding high-fat diets [84,85]. This association was also examined in humans by Gopinath et al. [86], which carried out a cross-sectional and longitudinal study to evaluate the relationship between dietary fats, certain food groups and SNHL, as well as to assess the association between serum lipids, the use of statins (cholesterol lowering drugs) and the progression of ARHL. This study found that high dietary intake of cholesterol was associated with an increased risk of developing SNHL. Among the mechanisms that could explain this association, disturbances of cochlear vasculature and atherosclerotic inflammatory changes, causing reduction of oxygen and nutrient supply together with decreased elimination of waste material, were mentioned. Neither dietary fat intake nor serum lipids showed any association with the incidence of HL. Hence, it was suggested that attention should be paid to the dietary patterns, specifically cholesterol intake, rather than serum lipid levels, as potential modifiable factors for the prevention of ARHL.

## 5. Micronutrients and Hearing Loss

### 5.1. Vitamins and Hearing Loss

The study of the effect of vitamins in HL has attracted the attention of numerous researchers. Vitamins, as micronutrients, have potential beneficial effects in the treatment/prevention of HL due to either their antioxidant properties or their essential role in the proper functioning of the ear. Several researchers have performed different animal or epidemiological studies to analyze the effect of these micronutrients in the auditory function (Table 3 and Table 4). Likewise, other studies have focused on the role of supplementation in the improvement of the auditory function (Table 5). Vitamins with antioxidant properties may prevent cochlear damage caused by high levels of toxic ROS produced e.g., during and after noise exposure. Therefore, most studies have focused on the preventive effect of their administration against noise-induced HL (NIHL), although some also focused on the ability of dietary antioxidants to prevent drug-induced ototoxicity [87]. Among others, chemicals such as glutathione, N-acetyl-L-cysteine, R-phenylisopropyladenosine and 2-oxothiazolidine-4-carboxylate were used to increase antioxidant levels. However, these approaches have important drawbacks regarding the drug administration route and the timing of multiple doses relative to noise exposure, so that the antioxidant is available in the cochlea at the time of ROS generation. Hence, dietary supplementation with antioxidant vitamins, such as vitamin C (ascorbate), E (α-tocopherol) and A (retinol), has been consider an alternative route to overcome these limitations. Therefore, different studies have deepened into their individual or combined effects in auditory function by means of the diet or the administration of supplements. For example, the effect of vitamin C supplementation in the prevention NIHL was evaluated by McFadden et al. in guinea pigs using three diets: normal, vitamin C deficient and supplemented diets [88]. No differences in ABR parameters were found among groups before noise exposure. However, post-exposure thresholds were approximately 15 dB lower in the supplemented group compared to the normal and deficient groups. Vitamin C supplementation also decreased significantly the permanent threshold shift in guinea pigs, whereas its deficiency had no effects on HL. These preliminary results offer interesting perspectives for development of practical approaches to prevent unavoidable damage caused by noise exposure, although additional studies using other models and vitamin C doses must be performed. Another study used Wistar albino rats to analyze prevention against cisplatin ototoxicity by a dose of vitamin B_1_, B_2_, B_6_, E, C or L-carnitine. ABR threshold and distortion product otoacoustic emissions were recorded 72 h after drug administration showing that vitamins and L-carnitine contributed to decrease or prevent cisplatin-induced ototoxicity, thus opening a very interesting pathway for its potential study and use in humans [87]. 

Since the antioxidant properties of vitamins A, C and E are exerted by different mechanisms, synergistic effects among dietary antioxidants and other dietary components can take place. As examples of different antioxidant mechanisms, vitamin E is a donor antioxidant in the cell membrane that reduces peroxyl radicals, inhibiting the propagation cycle of lipid peroxidation and vitamin C detoxifies free radicals in aqueous phase by means of their reduction [89,90]. Among other dietary components that can induce synergistic effects, magnesium should be considered. This mineral, mainly found in dried fruits and nuts and in legumes, reduces the formation of free radicals and thus noise-induced vasoconstriction [48]. Therefore, putative synergisms have been analyzed by Le Prell et al. using four groups of guinea pigs that received: (i) saline injections; (ii) a combination of vitamin A, C and E; (iii) only magnesium; and (iv) vitamins A, C and E plus magnesium. NIHL in animals that received the combined supplement of vitamins and magnesium was significantly reduced (lower ABR thresholds) than in the other groups. Thus, synergistic effects of vitamins A, C and E with magnesium were confirmed [91]. Furthermore, analysis of synergistic effects was extended using the C57BL/6 mice and six different antioxidants (L-cysteine-glutathione mixed disulphide, ribose- cysteine, NW-nitro-L-arginine methyl ester, vitamin B_12_, folate and ascorbic acid) that target four different sites within the oxidative pathway. After feeding the combined antioxidant therapy, a decrease in the threshold shift from baseline at all tested frequencies was detected compared to the control group [92]. This combination was also used in a longitudinal epidemiological study that analyzed the association between dietary antioxidants and ARHL in a cohort of Australian adults and evaluated its relation with the prevalence and 5-year incidence of HL [93]. Dietary data were analyzed using a food frequency questionnaire with 145 items that included reference portion sizes. Although this study demonstrated that higher intakes of vitamin A and E resulted in an inverse association with the prevalence of HL, it was not possible to demonstrate a reduction of the HL prevalence due to the intake of antioxidant combinations as shown in animal models. According to the results, dietary antioxidant consumption, alone or in combination, was unable to predict the 5-year incidence of ARHL. Another study evaluated a USA population aged 20–69 years using data from NHANES (National Health and Nutrition Examination Survey) 2001–2004, in which both food and dietary supplements were considered [94]. In contrast to animal studies, a dose-dependent trend between all individual nutrients and lower (better) speech pure tone average at both speech and high frequencies were reported, except for vitamin E. Moreover, this human study also provided evidence of the synergistic effect of high intakes of β-carotene and vitamin C with magnesium and a lower (better) pure tone average at high frequencies, and hence a reduced risk of HL. 

Remarkably, experimental models have demonstrated the importance of the genetic background in the dietary effects on HL progression. Dietary supplementation with β-carotene, vitamins C, E, and magnesium (ACEMg) to two animal models of hereditary deafness, the *Gjb2*-CKO (childhood deafness) and *Diap3*-Tg (auditory neuropathy), showed opposite results. The ACEMg diet slowed down the progression of HL in *Gjb2*-CKO mice and prompted a small but statistically significant improvement of auditory thresholds, probably owing to enhanced hair cell preservation. Conversely, in *Diap3*-Tg mice the diet had a detrimental effect, increasing HL compared to control fed animals [95]. Thus, these data suggest that only in certain backgrounds of genetic HL it is possible to modulate the phenotype by means of dietary supplementation. Importantly, *Diap3*-Tg mice present mutations in the *Cx26* gene, encoding connexin 26, which is the most common cause of SNHL. Lack of Connexin 26 triggers extracellular accumulation of potassium ions that results in an increase of glutamate concentrations, and hence oxidative stress [96]. These authors previously published a case-report where a boy with *Cx26*-related HL received ACEMg supplementation daily for three years and that, prior to the start of this regime, had a progressive worsening of the auditory function. As the supplementation began, audiometric evaluations confirmed no further progression of HL [97]. Of note, daily doses of each vitamin were higher than the recommended daily intake, but lower than the upper limit (15, 500 and 364 mg/day for vitamin A, C and E, respectively), whereas for magnesium the daily dose was lower than the recommended daily intake (167 mg/day). Therefore, it seems that these micronutrients play an essential role as antioxidants in the prevention of HL progression.

On the other hand, a Korean study performed in 2011 with more than 3200 subjects 50 to 80 years old, evaluated the effect of dietary vitamin intake and vitamin D serum concentrations in auditory function [98]. Carotenoid, retinol, thiamine, riboflavin, niacin and vitamin C intake were assessed by means of a 24-h recall. Evaluation of the auditory function included a physical examination of the ear to discard external or middle ear disease and a pure-tone audiometry to rule out conductive HL. High dietary intake of vitamin C was positively associated with a better auditory function in the midfrequency range. Hearing in the high and low frequencies was also improved in participants of the highest quartile of this vitamin. Regarding retinol, niacin and riboflavin consumption, only univariate linear regression analyzes showed significant correlation with better hearing at selective frequencies, indicating a minor association of these vitamins with HL as compared to factors such as age or noise exposure. Since vitamin consumption tends to decrease with age, the putative protection they offer will be reduced as life extends [99]. This study also presented association of high vitamin D serum concentrations and worse auditory function, a result that was in agreement with previous works that showed that vitamin D-deficient diets prevent ARHL in a mouse model [100] and that chronic sun exposure in the elderly could be a risk factor for HL [101]. 

The effect of the diet, by means of both nutrients and food group patterns, in the auditory threshold of a sample of French adults was also analyzed, as a continuation of a study that evaluated effects of dietary supplementation with antioxidant vitamins and minerals at nutritional doses on the incidence of cancer and ischemic cardiovascular disease [102]. At the end of the supplementation, some participants were included in an observational study to investigate the impact of nutrition in the quality of aging, including hearing function. Higher intakes of retinol and vitamin B_12_ were found to be associated with better auditory function only in women. Conversely, no association was found for β-carotene, folate and vitamins B_6_, C and E. Regarding food intake, women with higher consumption of meat as a whole, as well as red and organ meat, had better hearing levels compared to those with lower consumption of these food groups. For men, a higher consumption of seafood and shellfish improved auditory function [79]. This interesting study however has several limitations that must be taken into account, including the lack of consideration of confounders such as ototoxic medication, habitual noise exposure at work, medical conditions or genetic factors. In this line, a long-term study performed in the USA between 1991–2001, examined the association between intake of carotenoids, vitamins A, C and E and folate and the risk of HL in 65,521 women nurses [103]. Vitamin intake was assessed five times, with a gap of five years between each evaluation, using a food-frequency questionnaire. Interestingly, this study included covariates not analyzed in that of Péneau et al. such as hypertension or diabetes history and the consumption of potentially ototoxic medication such as acetaminophen or ibuprofen [79]. Inverse correlations between carotenoids (β-carotene and β-cryptoxanthin) and folate intakes with risk of acquired HL were observed. However, higher vitamin C intake from supplements and higher risk of HL were found, whereas no significant associations for vitamin A, E, and other carotenoids were detected [103]. 

Vitamin A, in the form of its active metabolite retinoic acid, is required for the normal development of the inner ear. In addition, in case of deficiency, malformations occur in a dose dependent-manner [104]. Vitamin A plays an essential role in the prevention of SNHL, especially during gestation, as showed by different studies [105,106]. Moreover, the existence of a synergism between vitamin A deficiency and infections is well known, as well as its relationship with increased risk of otitis media, whereas vitamin A supplementation could reduce the risk of HL due to otitis media. Based on this evidence, some authors consider that providing adequate amounts of vitamin A may reduce the risk of SNHL induced by its gestational deficiency [106]. To the best of our knowledge, only one study has been performed to date to elucidate if vitamin A supplementation improves auditory function. Preschool children received vitamin A supplementation, or a placebo capsule, every four months and their auditory function was assessed during their adolescence or early adulthood, sixteen years later [105]. In this setting, supplementation reduced the risk of HL associated with childhood ear infections. The authors postulated that this protective effect of vitamin A in HL may be due to an improvement of vitamin A-regulated defense systems, which entail maintenance of the epithelial integrity, modulation of oxidative stress and of the immune response, hence preventing all the inflammatory processes that cause hearing impairment [105]. However, important limitations of this study (i.e., only HL from middle ear infection was studied) indicate that further work is needed to elucidate if the dietary supplementation with this vitamin, either during gestation or in the early childhood, has a preventive role in HL either in animals or humans.

Vitamin B_12_ and FA deficiencies are the most common vitamin inadequacies in the elderly (jointly with vitamin D), and undoubtedly, the most studied in the auditory field. Vitamin B_12_ deficiency increases with age, mainly due to decreased production of acid and digestive enzymes needed to cleave the vitamin form found in food, and to the lack of intrinsic factor owing to the autoimmune destruction of gastric parietal cells caused by pernicious anaemia. According to NHANES data, the prevalence of vitamin B_12_ deficiency was estimated to be 2.9 to 25.7%, increasing with age and being generally higher in women than in men [107]. Vitamin B_12_ and FA play an important role in cellular metabolism (e.g., Hcy metabolism), the nervous system and vascular function, hence being also important in principle for the auditory function. Moreover, high Hcy concentrations, associated with low vitamin B_12_ or FA status or both, is a recognised risk factor for cerebral, coronary and peripheral vascular disease [108], hence affecting cochlear blood flow. A recently published review deeply analyzes Hcy metabolism and its relationship with cochlear function, its connection with SNHL, as well as its improvement by ω-3 supplementation [109]. In addition to Hcy serum concentrations, serum methylmalonic acid (MMA) is also useful to predict vitamin B_12_ deficits [107]. This parameter was used in a study of auditory function in older volunteers, showing that impaired hearing correlated with significantly higher MMA levels [110]. Additionally, vitamin B_12_ levels have been also related to other pathologies of the auditory system, such as tinnitus that is defined as the perception of sound in the absence of an external stimulus. In fact, a Turkish study detected that 63% of the patients with tinnitus had low B_12_ levels, whereas normal levels were found in a 37%, but no statically significant differences were observed as compared to the control group. Moreover, vitamin B_12_ replacement treatment was ineffective in improving the symptoms of tinnitus [111]. Considering the wide aetiology of tinnitus, further research is needed to find both, the causes and an effective treatment. 

The relationship between FA, Hcy and auditory function has been evaluated in different animal models. Kundu et al. [112] analyzed the effect of FA supplementation in drinking water in hyperhomocysteinemic mice (*Cbs^+/−^*) by means of a four-week treatment with a dose equivalent to 400 µg/70 kg/day and hearing thresholds were determined by ABR. Their findings suggested an improvement of hearing function through this supplementation, delineating, at the same time, a potential mechanism responsible for Hcy-associated-HL related to the levels of NOX subunits p22phox and p47phox and oxidative stress. Martínez-Vega el al. analyzed the auditory function in C57BL/6J mice, a mouse strain prone to HL, administering standard or folate-deficient diets for eight weeks. These authors observed that animals fed the FA deficient diet had an early onset of HL showing severe histological damage, impaired cochlear Hcy metabolism and oxidative stress [113]. These results were confirmed in a long-term study on the effects of FA deficiency using the CBA/J mouse strain with delayed HL onset, in which, again, the deficiency caused premature HL [114]. In both models, these results correlated with cochlear histological alterations. Differences observed in the initial signs of HL (2 months in C57BL/6J versus 8 months in CBA/J mice) indicate that the impact of the diet depends strongly on the genetic background. A recent pilot study analyzed the influence of different polymorphisms in genes of folate metabolism in the aetiology of presbycusis. Precisely, polymorphisms of 5,10-methylenetetrahydrofolate reductase (*MTHFR*), methionine synthase (*MTR,* catalysing Hcy remethylation) and thymidylate synthase (*TYMS*) on the onset of HL were evaluated. Remarkably, this study identified the association of variations in *MTHFR* and *TYMS* with ARHL, findings that could be of use for the prompt diagnosis of presbycusis [115].

Since both vitamin B_12_ and FA deficiency increase Hcy serum concentrations, most studies focused in the two vitamins together. Houston et al. were the first to evaluate the relationship between both vitamins and human auditory function in women aged 60–71 years [116]. Blood analyses, dietary records and auditory exams were performed to evaluate the influence of these vitamins and consistent associations were found between ARHL and low serum vitamin B_12_ and red cell folate. Since vitamin B_12_ and FA serum levels were highly correlated, it was impossible to identify whether any of these vitamins individually had a stronger association with pure-tone averages. Data from dietary analysis showed that auditory function presented a stronger association with folate intake rather than with vitamin B_12_ ingestion, the latter ascribed to B_12_ malabsorption due to the atrophic gastritis commonly found in the elderly. Lasisi et al. found that low serum levels of these two vitamins were significantly associated with HL at high frequencies in elderly volunteers [16]. At the same frequencies after adjusting for age, significant correlations were found with folate but not with B_12_. A cross-sectional Danish study evaluating 91 adults aged 67–88 years resulted in non-significant associations between serum folate, vitamin B_12_, Hcy and HL [117]. In order to clarify these contradictory results, a large prospective study in a representative old population cohort was carried out by Gopinath et al. for 5 years [118]. This study showed that nearly two-thirds of the people with high serum Hcy levels suffered from HL, whereas only one-third of those with normal serum Hcy levels presented this impairment. Additionally, low serum folate status increased 39% the risk of developing prevalent HL, whereas serum vitamin B_12_ levels had no association with ARHL. Regarding the 5-year incidence of HL, neither Hcy nor folate nor vitamin B_12_ serum levels showed predictive potential. The authors concluded that further studies with more participants and a longer follow-up are needed to evaluate the influence of Hcy and these vitamins as modifiable risk factors to reduce the AHRL incidence. A large prospective study, including more than 26,000 men with different health professions, aged 40–75 years at baseline in 1986, was performed and the follow up lasted until 2002. Their vitamin intake was assessed with a semi-quantitative food frequency questionnaire, as well as information on self-reported professionally diagnosed HL. This study reported no prospective associations between vitamin C, E, β-carotene or folate consumption and incident HL. However, higher folate intakes were associated with 21% reduction in the risk of HL in men aged from 60 years onward, but only observed in those volunteers with intakes ≥800 µg/day [119]. This value is considerably higher than the actual minimum recommendations for this vitamin, i.e., 400 µg/day for this age group. The main drawback of this study relies in the assessment of HL that was based on self-reported professionally diagnosed HL, probably underreporting other cases of HL. All these data should be considered for further studies, as well as the increased risk of folate malabsorption and depletion in the elderly. 

The relationship between low FA status and HL prompted Durga et al. to perform a double-blind, randomized, placebo-controlled trial to elucidate the effect of daily oral FA (800 µg) supplementation for 3 years [77]. Their results demonstrated that FA supplementation considerably slowed the decline of auditory function only at the speech frequencies, but not at high frequencies. Regarding the effect of vitamin B_12_ supplementation, a study conducted in the USA involving 93 volunteers aged 58–92 years was unable to demonstrate that this supplementation improved the hearing status in B_12_-deficient volunteers [110]. Despite the interest of these preliminary results, we can conclude that further studies should be performed considering the potential adverse effects of high doses of these vitamins. For example, results of a recently published study showed that adequate folate intake is beneficial for hearing, but a non-significant association with increased risk of HL was observed in volunteers with the higher folate intakes. The reason for this relationship is unclear, but the authors hypothesized that it could be related to the neurological damage associated to an excess of FA consumption [120]. Moreover, these adverse effects are thought to be worse in individuals with B_12_-deficiency [121]. Hereafter, supplementation with FA should be only administered to population subgroups with high risk of folate deficiency, to avoid the side effects of its excessive consumption. Therefore, these data reinforce the involvement of alterations in FA and Hcy metabolism in hearing disorders, although the mechanisms by which the cochlear function is affected remain poorly understood.

Finally, a recent study investigated the effect of nicotinamide or vitamin B_3_ in NIHL. This vitamin is a precursor of NAD^+^, a cofactor with an important role in the regulation of sirtuins. Animal studies demonstrated that administration of the vitamin, twice daily for 5 days before noise exposure and for 48 h thereafter, prevented NIHL [122]. Moreover, since the protective effect of nicotinamide in HL is related to its antioxidants properties [122], studies could be also extended to ARHL, therefore opening a new potential line of research in the field of nutrition and auditory function. 

### 5.2. Minerals and Hearing Loss

The role of minerals obtained either from dietary sources (Table 3 and Table 4) or supplements (Table 5) in the auditory function has been also addressed. Different studies have shown that low iron dietary intakes increase the risk of developing HL. Iron deficiency generates a subset of anaemia in which low hemoglobin, serum ferritin, serum iron, and/or increased soluble transferrin receptor are detected in patients, being oral iron supplementation its usual treatment. Among the pediatric population, premature infants, children exclusively fed with breast milk or formulas without iron fortification, children with reduced dietary intake or poor dietary absorption or those with significant blood loss show the highest risk of developing this anaemia. The connection between iron-deficiency anaemia and conductive and SNHL in a cohort of pediatric and adolescent volunteers aged 4–21 years was established by Schieffer et al. [127]. In their study, increased odds of SNHL were found in volunteers with iron deficiency anaemia, hence demonstrating the higher likelihood of these individuals to develop this kind of HL instead of conductive HL. The cochlea only receives blood from the labyrinthine artery, what explains its high susceptibility to the ischemic damage subsequent to iron deficiency anaemia [128]. Often, iron deficiency anaemia is secondary to malnutrition; hence the effect of protein-energy malnutrition and HL was studied in children aged 6–24 months. The data obtained by Kamel and co-workers indicated that 72% of the volunteers with moderate or severe protein energy malnutrition suffered from anaemia with hemoglobin levels of 9.51 ± 1.45 g/dL as compared to the control group. Moreover, a negative correlation between hemoglobin levels and ABR thresholds was observed, and therefore it was concluded that both protein energy malnutrition and anaemia are risk factors for hearing impairment [129]. 

The maternal iron nutritional state is crucial for the correct development of the offspring during both pregnancy and lactation. Although this mineral plays an important role in brain development and sensory maturation, including hearing, iron deficiency anaemia is commonly detected in pregnant women, since their increased demand is not fulfilled by the diet. The WHO estimated that this anaemia affected 38% of the pregnant women in 2011 [130]. Some studies have analyzed the impact of iron deficiency anaemia during pregnancy and lactation in auditory function in guinea pigs. Females received either iron sufficient or deficient diets during gestation, lactation and until the ninth postnatal day, thereafter all animals were fed the sufficient diet [30]. Higher ABR thresholds (worse acuity) were detected in females on the deficient diet at all tone frequencies, indicating that the anaemia affected different parts of the cochlea. The data also showed that iron-deficient pups suffered from SNHL according to the ABR latency-intensive curves, but whether this affectation is temporary or permanent could not be elucidated. However, it was suggested that even if HL disappears in young adulthood, these animals may have a higher risfor presbycusis or other morbidities during aging. The study by Yu et al. evaluated the effect of mild maternal iron deficiency anaemia, using the same regime than in the previous study, in cochlear function of the young guinea pigs, focusing on the putative role of apoptosis [123]. In this case, a significant decrease in the number of cochlear sensory hair cells due to an increase in the number of apoptotic cells was found in iron-deficient animals; this indicates the key role of apoptosis in inhibition of hair cell development, and results in a putatively alteration of the mechanisms underlying cochlear function. In a recent study, these same authors demonstrated that not only caspases (key molecules in the apoptosis cascade), but also other molecules such as prestin, glutamate transporter 3, and myosin VIIa have an important role in HL [123].

The influence of iron levels in HL is not only important in the pediatric population. Therefore, different studies have been carried out in Taiwanese [124] and American [131] adult populations that evaluate the relationship between iron deficiency anaemia and sudden SNHL, which is characterized by a rapid deterioration of hearing function that takes place in less than 72 h. Consistent results indicating a significant relationship between these parameters were found by Chung et al. [132] and Schieffer et al. [131]; noteworthy, SNHL was more severe in individuals under 60 years [124]. However, the underlying mechanism of this association is unclear, although it is thought to be caused by the exacerbation of vascular problems due to iron deficiency anaemia. 

Finally, different studies focused on the effect of iodine deficiency and thyroid function in HL [125,133,134,135]. The relationship between auditory and thyroid function is well known, but has not received much attention until now. A recent review critically analyzes the studies focused on this exciting topic and interested readers are referred to it [134]. Most of the work performed focused on the evaluation of hearing function in patients with congenital and acquired hypothyroidism, whereas only a few analyze the association between mild-to-moderate iodine deficiency and auditory function, especially in children. The population of Guizhou, a province of China with endemic iodine-deficiency, goitre, and cretinism, has poorer average auditory levels than other non-endemic areas making it an ideal subject for this type of studies. Three years of prophylaxis with iodine salt, showed an improvement of average hearing levels, reaching values similar to non-endemic controls, as well as recovery of normal thyroid function in the mean while [125]. Later on, Valeix et al. [133] reported more severe HL in children with mild-to-moderate iodine deficiency than in those with normal iodine levels and Azizi et al. [135] reported 44% and 15% prevalences in children with iodine deficiency as compared to 2% in children with adequate levels. These same authors found that dietary supplementation with iodized salt led towards a lower auditory threshold [135]. Hence, the limited data available reinforce the strong need for an in-depth investigation of the effect of iodine supplementation in auditory function at all life stages, including pregnancy, especially considering that thyroid malfunction is increasing lately in Europe. 

## 6. Other Dietary Factors and Hearing Loss

The analysis of single nutrients is not always possible and, therefore, different researchers opted to evaluate the overall dietary patterns in HL. Using data from the NHANES study, the relationship between the dietary quality measured with Healthy Eating Index (HEI) and the hearing sensitivity was studied in a population of adults aged 20 to 69 years. A statistically significant inverse correlation between the HEI and pure tone average at high frequencies was detected, being high HEI scores significantly associated with lower hearing thresholds at high frequencies. Although these results do not allow identification of causal relationships, they suggest that healthy eating strategies might benefit patients hearing function and overall life quality [136]. Nevertheless, additional prospective studies are needed for the evaluation of this relationship. 

The effect of other dietary components with antioxidant properties has been also investigated for ARHL, NIHL or drug-induced HL. For example, resveratrol, which is a phytoalexin produced by a wide variety of plants with numerous health benefits, including its prominent anti-aging effect, has been used in supplementation studies. Mice on a high caloric diet, when supplemented with resveratrol, extend their lifespan and increase their mitochondrial biogenesis [137]; in this anti-aging effect activation of the sirtuin pathway has a key role. In the auditory system, resveratrol effectively reduced NIHL in F344/NHsd [138] and Wistar albino rats [139], and, additionally, it was reported to protected against cisplatin-dependent ototoxicity, a very interesting way to counteract the adverse effects of this chemotherapy [140]. Other antioxidants, such as caroverine [141] or N-acetyl-L-cysteine [142,143] also have protective effects against NIHL in rats and chinchillas. de Rivera et al. evaluated the effect that a blueberry-enriched diet has in age-related decline of auditory system. Specifically, they used frequency modulated sweeps, which are characterized by changes in frequency over time, in the primary auditory cortex. Aged rats fed blueberry-enriched diet showed faster frequency modulated sweeps, similar than those of young rats, hence confirming the potential of blueberries to reverse ARHL [144].

## 7. Conclusions

The available results suggest that the nutritional status plays an important role in the maintenance of the auditory capacity. Caloric restriction may be an interesting option to avoid HL, as suggested by results obtained in animal models, but extensive human studies are needed to verify this possibility. Additionally, there is an increasing number of data supporting the idea that the concentration of essential macronutrients and micronutrients in our diet has a high impact on the occurrence and progression of HL. The available evidences show that nutritional interventions can prevent, at least partially, this sensorial deficit. In this line, ω3 fatty acids or dietary antioxidants may be potentially useful as deduced from several epidemiological and animal studies. Nevertheless, the animal models analyzed, and the longitudinal studies carried out to date are still limited to obtain meaningful conclusions. Likewise, studies devoted to the assessment of supplementation with micronutrients, such as vitamins or certain minerals, have provided promising results. Vitamins A and folic acid, as well as minerals such as iodine may offer an alternative to improve HL in a population at risk by slowing down the progression of the disorder as deduced from the available studies carried to date. Moreover, data derived specially from animal models have revealed the importance of the genetic background in the outcomes obtained from different dietary patterns and some associations to single nucleotide polymorphisms are emerging in addition to the well-known role of connections in hearing impairments. 

Altogether, we can conclude that the efforts carried out to date to uncover the mechanisms involved in the interplay between nutrition and HL have not been in vain, despite the limited, and sometimes contradictory, results available. However, there is a need to deepen into this relationship, particularly as HL is becoming a problem of great magnitude for western countries, due to the demographic change they are experiencing (a substantial increase in their elderly populations). This fact is turning to be of great economic impact, both for the society and the individual, and hence the implementation of preventive protocols, in parallel to the development of effective therapies, seems crucial to improve the quality of life of the whole society. 

## Figures and Tables

**Figure 1 nutrients-11-00035-f001:**
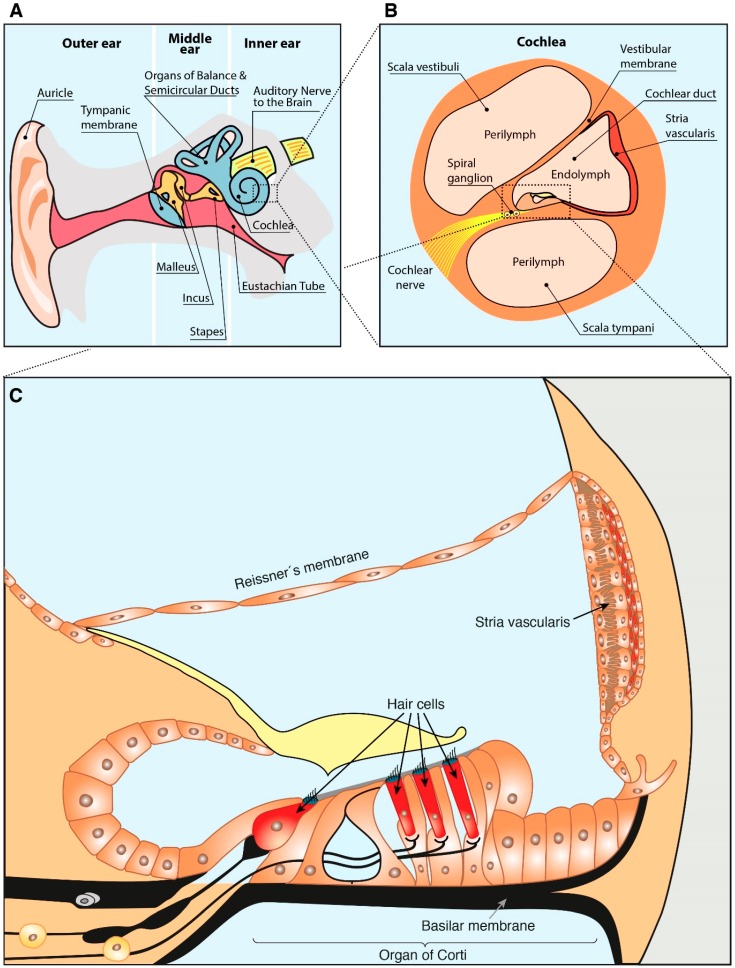
Schematic drawing of the ear anatomy. (**A**) Ear anatomical compartments. (**B**) Cochlea anatomy. (**C**) Structure of the cochlear scala media. Based on Sanchez-Calderon et al. [25] and Rivera T. et al. [26].

**Table 1 nutrients-11-00035-t001:** Summary of the effects of caloric restriction in the auditory function.

Animal Model	Effect	Author and Year
Sprague–Dawley albino rats	Maintenance of the auditory reflex and cellular integrity of the stria vascularis	Mannström et al., 2013 [57]
C57BL/6J mice	Cochlear degeneration reduction and Sirt3 induction in the cochlea	Someya et al., 2007 [60]
Rhesus monkeys	Lower ABR threshold compared than control	Fowler et al., 2002; Ramsey et al., 2000 [62,63]
Rhesus monkeys	No differences with control	Torre et al., 2004 [64]

**Table 2 nutrients-11-00035-t002:** Main results of research studies that evaluate the effect of macronutrients on the auditory function.

Macronutrient	Study Subjects/Animal Model	Effect	Author and Year
Carbohydrates	Adults (humans)	High glycemic index, glycemic load and overall total carbohydrate levels increase HL risk	Gopinath et al., 2010 [68]
Proteins	C57BL/6J wild-type mice	High β-Conglycinin consumption prevent the increase of the ABR threshold	Tanigawa et al., 2015 [70]
Lipids	Wistar rats	Animals with high ω-3 consumption suffer from permanent raises in ABR thresholds	Church et al., 2010 [80]
	C57BL/6J mice	Preventive effect of the ω-3-supplemented diet in the loss of hearing acuity	Martinez-Vega et al., 2015 [83]
	Guinea pigs and chinchillas	High dietary saturated and monounsaturated fats and cholesterol intake increase hearing threshold	Saito et al., 1986; Sikora et al., 1986 [84,85]
	Adults (humans)	Significant inverse associated between dietary total long chain ω-3 in plasma and the prevalence of HL	Gopinath et al., 2010 [21]
		Negative association between regular ingestion of fish and 5-years incidence and progression of HL	
	Adults (humans)	Two or more fish servings consumption per week reduced HL	Curhan et al., 2014 [78]
		High long-chain ω-3 intake was inversely associated with HL risk	
	Adults (humans)	No association between fish consumption (with ω-3) and HL	Péneau et al., 2013 [79]
	Adults (humans)	High dietary intake of cholesterol was associated with an increased risk SNHL	Gopinath et al., 2011 [86]
		Neither dietary fat intake nor serum lipids showed any association with HL incidence	

**Table 3 nutrients-11-00035-t003:** Main results of studies with animal models that evaluate the effect of micronutrients in the auditory function. HL: hearing loss; FA: folic acid; SNHL: sensorineural HL; ARHL: age-related HL.

Micronutrient	Animal Model	Effect	Author and Year
Vitamin C	Guinea pigs	Supplementation significantly decreased the permanent ABR threshold shift whereas deficiency had no effects on NIHL	McFadden et al., 2005 [88]
Vitamins B1, B2, B6, E, C	Wistar albino rats	Vitamins contributed to decrease or prevent cisplatin-induced ototoxicity	Tokgöz SA, et al., 2012 [87]
Vitamin A, C, E and magnesium	Guinea pigs	Synergistic effects of vitamins A, C and E with magnesium in the reduction of noise-induced threshold	Le Prell et al., 2007 [91]
Dietary antioxidants	C57BL/6 mice	Synergistic effect of the antioxidants decreasing the threshold shift from baseline at all frequencies compared to control group	Heman-Ackah et al., 2010 [92]
β-carotene, vitamins C, E, and magnesium	Mice model of hereditary deafness (Gjb2-CKO and Diap3-Tg)	Dietary supplementation slowed down the progression of HL and improved auditory thresholds in Gjb2-CKO mice	Green et al., 2016 [95]
		HL increase in Diap3-Tg mice compared to control	
Folic acid	Cbs+/− Mice	FA supplementation in hyperhomocysteinemic mice led to improvement of hearing function	Kundu et al., 2012 [112]
Folic acid	C57BL/6J mice	HL detected after two-months on a FA deficient diet	Martinez-Vega el al., 2015 [113]
		Correlation between HL, hyperhomocysteinemia and histological damage in the cochleae	
Folic acid	CBA/J mice	Initial signs of HL detected after 8-months of vitamin deficiency	Martinez-Vega el al., 2016 [114].
		Correlation between HL, hyperhomocysteinemia and histological damage in the cochleae	
Nicotinamide riboside	C57BL/6 mice	Nicotinamide riboside administration, twice daily for 5 days before noise exposure and for 48 h thereafter, prevented NIHL	Brown et al., 2014 [122]
Iron	Guinea pigs	Higher ABR thresholds (worse acuity) were detected in females fed with iron deficient diet	Jougleux et al., 2011 [30]
		Iron-deficient pups suffered from SNHL	

**Table 4 nutrients-11-00035-t004:** Main results of the epidemiological studies that evaluate micronutrients effects on hearing loss.

Micronutrient	Study Subjects	Effect	Author and Year
Carotenoid, retinol, thiamine, riboflavin, niacin and vitamin C	Adults	High dietary intake of vitamin C was associated with a better auditory function	Kang et al., 2014 [98]
		Retinol, niacin and riboflavin consumption showed minor association with HL	
		Vitamin D serum concentration associated with worse auditory function	
Retinol, vitamin B_12_, β-carotene, folate, vitamins B6, C and E	Adults	High retinol and vitamin B_12_ intake associated with better auditory function in women	Hercberg et al., 2004 [102]
		No associations found for β-carotene, folate and vitamins B6, C and E	
Carotenoids, FA, vitamins C, A and E	Women	Inverse correlations between carotenoids (β-carotene and β-cryptoxanthin) and folate intakes and risk of acquired HL	Curhan et al., 2015 [103]
		Direct correlation between high vitamin C intake (from supplements) and risk of HL	
		No significant associations for vitamin A, E, and other carotenoids	
Dietary antioxidants	Adults	High vitamin A and E consumption showed inverse associations with HL prevalence	Gopinath et al., 2011 [93]
		Dietary antioxidants consumed alone or in combination were unable to predict 5-year incidence of ARHL	
Vitamins and minerals	Adults	Dose-dependent trend between all individual nutrients (except vitamin E) and better speech pure tone average	Choi et al.,2014 [94]
		Synergistic effect of high intakes of β-carotene and vitamin C with magnesium and better pure tone average at high frequencies	
Folic acid and vitamin B_12_	Women	Consistent associations between low vitamin B_12_ and folate levels and ARHL	Houston et al., 1999 [112]
		Stronger association with folate intake rather than with vitamin B_12_ ingestion, according to dietary intake	
Folic acid and vitamin B_12_	Elderly	Low serum vitamin levels were significantly associated with HL in the high frequencies	Lasisi et al., 2016 [16]
		Significant correlations, after adjusting for age, in folate but not in B12	
Folic acid and vitamin B_12_	Adults	Non-significant associations between serum folate, vitamin B_12_, Hcy and HL	Berner et al., 2000 [113]
Folic acid and vitamin B_12_	Adults	Low serum folate status increased risk of developing HL	Gopinath et al., 2010 [114]
		Serum vitamin B_12_ levels had no association with ARHL	
		Neither folate nor vitamin B_12_ showed predictive potential for 5-year incidence of HL	
Folic acid, carotene, vitamins C and E	Men	No prospective associations between vitamin C, E, β-carotene or folate consumption and HL	Shargorodsky et al., 2010 [119]
		High folate intakes were associated with reduced risk of HL	
Folic acid	Adults	Adequate folate intake is beneficial for hearing	Kabagambe et al., 2018 [116]
		High folate intake showed non-significant association with an increased risk of HL	
Iron	Children and adolescents	Increased odds of SNHL in volunteers with iron deficiency anaemia	Schieefer et al., 2017 [119]
Iron	Children	Negative correlation between hemoglobin levels and auditory function	Kamel et al., 2016 [121]
Iodine	Children	More severe HL in children with mild-to-moderate iodine deficiency compared to those with normal iodine levels	Valeix et al., 1994 [123]
Iodine	Children	HL prevalences of 44 to 15% in children with iodine deficiency compared to 2% in children with adequate levels	Azizi et al., 1993 [124]

**Table 5 nutrients-11-00035-t005:** Intervention studies to analyze supplementation effect on hearing loss.

Micronutrient	Study Subjects	Effect	Author and Year
Vitamin A	Preschool children	Vitamin A supplementation reduced risk of HL associated with childhood ear infections	Schmitz et al., 2012 [105]
Folic acid	Adults	Folic acid (800 µg) supplementation for 3 years slowed the decline of auditory function only at speech frequencies, but not at high frequencies	Durga et al., 2007 [77]
β-carotene, vitamins C, E, and magnesium	Boy with *Cx26*-related HL	Daily supplementation led to no further progression of HL	Thatcher et al., 2014 [97]
Vitamin B_12_	Adults	Not possible to demonstrate an improvement on hearing status with vitamin B_12_ supplementation	Park et al., 2006 [110]
Iodine	Adults	Prophylaxis with iodine salt (three years) revealed an improvement of average hearing levels, reaching values similar to controls	Wang and Yang.1985 [125]
Iodine	Children	Iodine dietary supplementation with ionised salt led towards a lower auditory threshold	Azizi et al.,1993 [126]

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
