# Peer review of "Interplay between Nutrition and Hearing Loss: State of Art"

_nutrients, 2018, doi:10.3390/nu11010035_

Reviewer 1 Report

General comments

In this review paper, the authors summarise evidence for the effects of nutrients on auditory function. In the main, the review is reasonably clear and well-written, however there are certain places where the text needs to be clarified (see specific comments below). The paper will be of interest to readers of Nutrients. I have several comments that the authors may wish to address in order to strengthen the manuscript.

 The title is “Nutrition and Hearing Loss: New Challenges and Opportunities.” While the paper provides a good review of how nutrition impacts on hearing loss, the new challenges and opportunities are never really made clear. Perhaps this could be addressed in a revised version.

 An area for possible improvement is the structure and layout of the review. Currently, the review is composed entirely of an Introduction (section 1, with many subheadings), and a Conclusions section (section 2), which is just a single paragraph at the end. Inclusion of more sections should be considered. Indeed, many of the subheadings appear to be separate sections in and of themselves and there does not appear to be any reason why they are part of the Introduction (e.g. 1.2. Caloric restriction and hearing loss – why is this not section 2?). The Introduction section itself needs to end sooner, with a closing description of the contents of the following review that allows the reader to clearly see how the rest of the paper will be laid out.  

 In addition, the conclusions section feels rushed and is not particularly helpful in its current format. It could be significantly improved by summarising the main findings and conclusions to be drawn from the evidence presented throughout the review. Suggestions for future research, and highlighting gaps in current knowledge, would also strengthen this section.

 Specific comments

Line 58: “For those who develop HL, early diagnosis and appropriate interventions are adequate to achieve improved outcomes.” This is rather vague. Please briefly expand on this statement and provide a supporting citation.

 Line 66 (or thereabouts): The Introduction provides a good overview of the prevalence of hearing loss, the consequences for quality of life of hearing loss, and a discussion of how the prevalence and impact of HL can be mitigated through public health actions. What is missing is an overview of how the review will be structured and the content of the rest of the paper. Inclusion of a paragraph outlining this would help the reader.

 Line 135: “important for distinguishing deaf consonants…” Please state what a “deaf” consonant is.

 Line 387: “supplementation with antioxidants has been consider an alternative route…” Suggest “…considered…”

 Line 468: “…worse auditory function, result that was in…” Suggest “…worse auditory function, a result that was in…”

 Line 509: “i.e. it only studied a specific kind of HL”. Please state what kind of HL is being referred to here.

 Line 526: “This parameter was used in a study of auditory function in ancient volunteers…” Ancient volunteers? Are the authors referring to “older” volunteers? Suggest rewording this.

 Line 531: “levels were found in a 37%, but no statically significant differences were observed among those groups.” What differences are being referred to here? Please be more specific or reword to make this clearer.

 Line 680: “A few studies have directly investigated the effect of iodine supplementation on hearing function. These same authors…” Please provide citations of these studies. What “same authors” are being referred to here?

 Line 682: “Although the limited data suggest that iodine deficiency is related to HL, data are quite old and not sufficient to evidence causality.” I do not understand this. What is meant by “data are quite old”, and why is this a problem? Why are the data not sufficient to evidence causality?

Author Response

Reviewer 1

 General comments

In this review paper, the authors summarize evidence for the effects of nutrients on auditory function. In the main, the review is reasonably clear and well-written, however there are certain places where the text needs to be clarified (see specific comments below). The paper will be of interest to readers of Nutrients. I have several comments that the authors may wish to address in order to strengthen the manuscript.

The title is “Nutrition and Hearing Loss: New Challenges and Opportunities.” While the paper provides a good review of how nutrition impacts on hearing loss, the new challenges and opportunities are never really made clear. Perhaps this could be addressed in a revised version.

We sincerely appreciate the comment of the reviewer and his/her suggestion to improve our manuscript. Accordingly, the title of the manuscript has been changed to “Interrelation between Nutrition and Hearing Loss: State of Art”.

An area for possible improvement is the structure and layout of the review. Currently, the review is composed entirely of an Introduction (section 1, with many subheadings), and a Conclusions section (section 2), which is just a single paragraph at the end. Inclusion of more sections should be considered. Indeed, many of the subheadings appear to be separate sections in and of themselves and there does not appear to be any reason why they are part of the Introduction (e.g. 1.2. Caloric restriction and hearing loss – why is this not section 2?). The Introduction section itself needs to end sooner, with a closing description of the contents of the following review that allows the reader to clearly see how the rest of the paper will be laid out.  

In line with the kind comments of the reviewer, we have now included different sections and a brief description of the contents at the end of the Introduction section for a better comprehension by the reader.

In addition, the conclusions section feels rushed and is not particularly helpful in its current format. It could be significantly improved by summarising the main findings and conclusions to be drawn from the evidence presented throughout the review. Suggestions for future research, and highlighting gaps in current knowledge, would also strengthen this section.

Following the reviewer’s suggestion, the Conclusions section has been rewritten according to his/her recommendations.

 Specific comments

Line 58: “For those who develop HL, early diagnosis and appropriate interventions are adequate to achieve improved outcomes.” This is rather vague. Please briefly expand on this statement and provide a supporting citation.

The suggested change has been included in the new version of the manuscript.

Line 66 (or thereabouts): The Introduction provides a good overview of the prevalence of hearing loss, the consequences for quality of life of hearing loss, and a discussion of how the prevalence and impact of HL can be mitigated through public health actions. What is missing is an overview of how the review will be structured and the content of the rest of the paper. Inclusion of a paragraph outlining this would help the reader.

Thank you for your comments. As indicated before, a brief explanation about the structure of the paper (Lines 68-73) has been included in the revised version of the manuscript.

Line 135: “important for distinguishing deaf consonants…” Please state what a “deaf” consonant is.

Following your suggestion, a statement explaining the meaning of deaf or voiceless consonants, as well as examples of those letters are now included (Line 158-159). 

Line 387: “supplementation with antioxidants has been consider an alternative route…” Suggest “…considered…”

As suggested by the reviewer, the verbal tense has been changed.  

Line 468: “…worse auditory function, result that was in…” Suggest “…worse auditory function, a result that was in…”

The text has been modified (Line 502), following your kind suggestion.

Line 509: “i.e. it only studied a specific kind of HL”. Please state what kind of HL is being referred to here.

The suggested change has been introduced.

Line 526: “This parameter was used in a study of auditory function in ancient volunteers…” Ancient volunteers? Are the authors referring to “older” volunteers? Suggest rewording this.

According to your suggestion, we have substituted “ancient” for “older”

Line 531: “levels were found in a 37%, but no statically significant differences were observed among those groups.” What differences are being referred to here? Please be more specific or reword to make this clearer.

Also, attending to your proposal, we have added the suggested clarification.

Line 680: “A few studies have directly investigated the effect of iodine supplementation on hearing function. These same authors…” Please provide citations of these studies. What “same authors” are being referred to here?

As recommended, sentences referring to this study have been rephrased and the appropriate citation is now included.

Line 682: “Although the limited data suggest that iodine deficiency is related to HL, data are quite old and not sufficient to evidence causality.” I do not understand this. What is meant by “data are quite old”, and why is this a problem? Why are the data not sufficient to evidence causality?

Thank you very much for your comment, highlighting this problem. The corresponding sentence has been rewritten in the new version of the manuscript to avoid misunderstandings.

Reviewer 2 Report

Content

There are a few areas of unclear or incorrect information that needs to be revised:

 1. Line 53 is an incomplete sentence; please specify what types of clinical trials are currently under research in U.S.

 2. The term “neurosensorial hearing loss (NSHL)”, first appearing on lines 82/83 is not in English usage; this should be substituted with “sensorineural hearing loss (SNHL)”, and replaced throughout the paper.

3. The categories of hearing loss are listed incorrectly in the paper (lines 84-88), as is the citation; this should be corrected. See information and source for the ASHA citation below.

 https://www.asha.org/public/hearing/Degree-of-Hearing-Loss/

Degree of hearing loss

Hearing loss range (dB HL)

Normal

–10 to 15

Slight

16 to 25

Mild

26 to 40

Moderate

41 to 55

Moderately severe

56 to 70

Severe

71 to 90

Profound

91+

Source: Clark, J. G. (1981). Uses and abuses of hearing loss   classification. Asha, 23, 493–500.

4. Lines 92-96; reference number 29 is not an appropriate source for this information.

 5. Lines 110-112 “For example, white and…”; please state which study are you referencing here;

 6. Lines 114-118; which study is being referenced here? Please state authors

 7. Consider moving the section on review of ear anatomy into a separate section following the Introduction and move the purpose statement (line 144) to the end of the Introduction section.

 8. Line 135: what is meant by “deaf” consonants? Please rephrase.

9. Please rephrase statement on lines 137 “In addition, people with ARHL have difficulties in locating the source of the sound…”; please note that difficulty with localization occurs primarily in asymmetrical hearing loss and is not a uniform characteristic of SNHL.                                         

10. Please check statement on lines 137-8 “HL progresses by approximately 1dB per year, once 60 years of age are reached”; this could not be verified in the reference:

Ciorba A; Bianchini C; Pelucchi, S.; Pastore, A. The impact of hearing loss on the quality of life of elderly adults. Clin Interv Aging 2012, 7, 159-163.

 11. Lines 181 and 190, where the terms “These researchers”, and “They observed”.. are used, please specify which researchers are referenced.

 12. Line 471, where it is stated “…in the auditory function…”, please specify which aspect of auditory function is referenced here; is it threshold?

 13. Line 502: “Preschool children…”; please cite the study that is being discussed here.

 14. Line 631: in sentence beginning with “The data…”; the specific authors of this study should be cited here.

 15. Line 634: specify what is meant by “auditory function”; was this a change in hearing thresholds? Please clarify.

 16. Lines 684-686 beginning with “Moreover…” needs a citation.

 17. Lines 689: referencing the NHANES study, please indicate which population is being discussed.

 18. Line 708: “Faster frequency modulated sweeps…”, please clarify which study is being referenced and which measure is described.

 19. The Conclusions section is weak and needs to be rewritten.

Writing Mechanics

 Several sections of the manuscript need to be re-written for clarity:

1.     Lines 33-35 beginning with ‘Among”..please re-write; awkward sentence structure.

 2.     Line 35: sentence beginning with: “Therefore, this is one of the fields…”; specify, what is meant by “this”.

 3.     Lines 42-45, beginning with “Worsening of…” is an awkward sentence; please re-write.

 4.     Re-write line 46 “people’s life”…

 5.     Re-Write lines 46-49; clarify what is meant by the “reported effects”; also “blaming, withdrawing, or bluffing”…

       6. Line 387: should be written “…has been considered”…

      7. Line 526: word “ancient” should be replaced with “older adults”.

      8. Line 624: word “connexion” is misspelled; should be “connection”.

      9. Line 629: the word “what” should be replaced with “which”.

      10. Line 669: remove the colloquial word “exciting”.

      11. Lines 673-680 beginning with “The population of Guizhou…” should be rewritten for clarity.

Author Response

Reviewer 2

There are a few areas of unclear or incorrect information that needs to be revised:

 1. Line 53 is an incomplete sentence; please specify what types of clinical trials are currently under research in U.S.

We sincerely appreciate the comments of the reviewer and his/her new suggestion to improve our manuscript. Accordingly, the sentence has been completed with the required information (Line 55-57).

2. The term “neurosensorial hearing loss (NSHL)”, first appearing on lines 82/83 is not in English usage; this should be substituted with “sensorineural hearing loss (SNHL)”, and replaced throughout the paper.

 The mistake kindly noticed by the reviewer has been corrected in the new version of the manuscript.

 3. The categories of hearing loss are listed incorrectly in the paper (lines 84-88), as is the citation; this should be corrected. See information and source for the ASHA citation below.

 https://www.asha.org/public/hearing/Degree-of-Hearing-Loss/

  The new version of the manuscript now includes the data of the ASHA citation kindly suggested by the reviewer.

 4. Lines 92-96; reference number 29 is not an appropriate source for this information.

 According to your suggestion, reference 29 has been substituted by reference 25 (Cediel R, et al., Sensorineural hearing loss in insulin-like growth factor I-null mice: a new model of human deafness. Eur J Neurosci, 2006. 23(2): p. 587-90..), which is more accurate.

 5. Lines 110-112 “For example, white and…”; please state which study are you referencing here;

 Thank you for your observation. We have included in the text the reference of the study we mention (Line 134). 

 6. Lines 114-118; which study is being referenced here? Please state authors

 We have included the specific references.

 7. Consider moving the section on review of ear anatomy into a separate section following the Introduction and move the purpose statement (line 144) to the end of the Introduction section.

 According to the reviewer’s suggestion, we have included the requested new ear anatomy section following Introduction (Lines 75-83). Moreover, the purpose statement has been moved to the end of the Introduction section (Lines 68-73).

8. Line 135: what is meant by “deaf” consonants? Please rephrase.

Thank you very much for your suggestion. A statement explaining the meaning of deaf or voiceless consonants, and examples of those letters, are included in the new version of the manuscript (Line 158-159). 

9. Please rephrase statement on lines 137 “In addition, people with ARHL have difficulties in locating the source of the sound…”; please note that difficulty with localization occurs primarily in asymmetrical hearing loss and is not a uniform characteristic of SNHL.

This mistake was corrected in the new version of the manuscript. Thank you.

10. Please check statement on lines 137-8 “HL progresses by approximately 1dB per year, once 60 years of age are reached”; this could not be verified in the reference:

Ciorba A; Bianchini C; Pelucchi, S.; Pastore, A. The impact of hearing loss on the quality of life of elderly adults. Clin Interv Aging 2012, 7, 159-163.

We have replaced this reference by a more appropriate citation in the new version of manuscript.

11. Lines 181 and 190, where the terms “These researchers”, and “They observed”.. are used, please specify which researchers are referenced.

According to your suggestion, researcher´s names are now included in the manuscript.

12. Line 471, where it is stated “…in the auditory function…”, please specify which aspect of auditory function is referenced here; is it threshold?

As kindly requested by the reviewer, we have included in the new version of the manuscript that the effect of the diet on the auditory function was evaluated by means of the analysis of the auditory thresholds.

13. Line 502: “Preschool children…”; please cite the study that is being discussed here.

The reference was included in the new version of the manuscript.

14. Line 631: in sentence beginning with “The data…”; the specific authors of this study should be cited here.

The names of the authors of this study has been now included. 

15. Line 634: specify what is meant by “auditory function”; was this a change in hearing thresholds? Please clarify.

Following the reviewer´s suggestion, we have modified the text clarifying that negatives correlations were found between haemoglobin levels and ABR thresholds.

16. Lines 684-686 beginning with “Moreover…” needs a citation.

The requested citation is now included in the manuscript.

17. Lines 689: referencing the NHANES study, please indicate which population is being discussed.

 The manuscript includes now the characteristics of the population of the study discussed. Thank you.

18. Line 708: “Faster frequency modulated sweeps…”, please clarify which study is being referenced and which measure is described.

According to your recommendations, we have included a brief description of the technique used in the study to evaluate the auditory function and we have clarified the obtained results in the text (Lines 761-766).

19. The Conclusions section is weak and needs to be rewritten.

The conclusions section has been rewritten according to the reviewer’s suggestion.

Round  2

Reviewer 1 Report

The manuscript is improved from the previous version, and the authors have addressed the comments from the previous round of review. My only comment that the authors may wish to consider is changing the new title from “Interplay between Nutrition and Hearing Loss: State of Art” to “Interplay between Nutrition and Hearing Loss,” or similar (such as “Nutrition and hearing loss”), as “State of Art” is confusing in the title and it is unclear what is being referred to. Otherwise, my recommendation is accept.